# Get Your Guidance Going: Investigating the Activation of Spatial Priors for Efficient Search in Virtual Reality

**DOI:** 10.3390/brainsci11010044

**Published:** 2021-01-04

**Authors:** Julia Beitner, Jason Helbing, Dejan Draschkow, Melissa L.-H. Võ

**Affiliations:** 1Scene Grammar Lab, Institute of Psychology, Goethe University, 60323 Frankfurt am Main, Germany; jason.helbing@stud.uni-frankfurt.de (J.H.); mlvo@psych.uni-frankfurt.de (M.L.-H.V.); 2Brain and Cognition Laboratory, Department of Psychiatry, University of Oxford, Oxford OX3 7JX, UK; dejan.draschkow@psych.ox.ac.uk

**Keywords:** visual search, repeated search, incidental memory, virtual reality

## Abstract

Repeated search studies are a hallmark in the investigation of the interplay between memory and attention. Due to a usually employed averaging, a substantial decrease in response times occurring between the first and second search through the same search environment is rarely discussed. This search initiation effect is often the most dramatic decrease in search times in a series of sequential searches. The nature of this initial lack of search efficiency has thus far remained unexplored. We tested the hypothesis that the activation of spatial priors leads to this search efficiency profile. Before searching repeatedly through scenes in VR, participants either (1) previewed the scene, (2) saw an interrupted preview, or (3) started searching immediately. The search initiation effect was present in the latter condition but in neither of the preview conditions. Eye movement metrics revealed that the locus of this effect lies in search guidance instead of search initiation or decision time, and was beyond effects of object learning or incidental memory. Our study suggests that upon visual processing of an environment, a process of activating spatial priors to enable orientation is initiated, which takes a toll on search time at first, but once activated it can be used to guide subsequent searches.

## 1. Introduction

Imagine you are invited to your new colleague’s home to prepare dinner together. Even though you see this kitchen for the first time, you already have an idea about where to find the ingredients you need for the spaghetti with tomato sauce. You look for the pasta and tomatoes in the storage rack. Then, you find some basil on the windowsill and get a pot out of the cupboard. Tasks like these are an essential part of our daily routines and while computationally challenging for artificial systems, these tasks seem effortless to us [1,2,3]. A central reason for the efficiency of these processes in humans is the availability of rich prior knowledge, which can help guide behavior [3,4,5,6,7]. But which priors are activated upon searching and how do they influence subsequent search guidance?

A powerful and popular method of assessing the contributions of priors on behavioral performance is repeated visual search, i.e., letting observers search for several objects successively in the same environment. While priors are important for alleviating capacity-limited cognitive processes (like the allocation of spatial attention), this guidance might not always result in behavioral advantages. When using two-dimensional photographs of scenes the time it takes us to find a target remains constant over repeatedly searching through an unchanging visual scene [5,8,9], implying that memory from previous encounters is not used to speed subsequent searches. Even though memories of object identities and locations are acquired while repeatedly searching [10,11,12,13,14], search efficiency can remain static after more than 250 searches [15]. Overall, the evidence for and against the use of memory in repeated search has been mixed (for a review see [5]). This may be due to greatly varying search displays used across studies, ranging from artificial letter displays [15,16,17,18,19,20] to real-world scenes on screen [8,9,21,22,23,24], in virtual reality (VR) [12,25], or actual real-world environments [11].

While repeated search has proven to be a powerful paradigm for the investigation of how priors guide attentional allocation, most studies average response times across multiple consecutive trials into epochs. Thus, there has been little direct evidence for memory utilization from one search to the next. When looking at individual trials, Draschkow, Stänicke, and Võ [26] made the observation that the first search of several subsequent searches in a stimulus array or scene always took the longest. When comparing the first search and all subsequent searches they found this “search initiation effect” (SIE) consistently over various studies with different designs, stimuli, and manipulations [8,14,15,16,17,25,27].

In fact, Wolfe, Klempen, and Dahlen [15] provided one of the only reports granular enough to demonstrate an observable drop in search time from the first to the second trial but not to the subsequent. In a series of experiments, participants had to repeatedly search through the same array of letters and shapes and the stimulus onset asynchrony was varied. The search time difference between the first two trials within the same array was explained by the additional cognitive processing load induced by a simultaneous onset of the target cue and the search display and was not further analyzed. When varying the stimulus onset asynchrony, i.e., presenting the search cue before or after the onset of the search display, this increase in response time disappeared. This observation could be explained by the increased demand for coincident processing of search array and cue. Switches between attention to the cue and the array might cause delays in processing. This brief mental overload might result in longer processing times thereby increasing response times.

Indeed, the first search within a scene has some qualitatively different demands than the subsequent searches. It (often) involves the first visual processing of the scene which elicits a cascade of mental processes such as gist processing [28,29,30], scene categorization [31,32,33], and spatial prior activation [31,34]. Visual search studies using previews of the upcoming stimulus context have shown that scene previews lead to a significant reduction in subsequent search times [21,28,35] with preview durations as brief as 250 ms [36] or even 50 ms [29]. The implicit information learned through previews is then used to, for example, efficiently guide eye movements to probable target locations [28,30,35]. This observation has been coined the preview effect [13,21] or preview benefit [28,35,36]. In earlier research, the preview effect has been mainly analyzed by comparing search times and eye movement behavior after seeing a preview compared to searches without a preview. Despite finding search time differences, studies investigating the preview effect typically look at one search per scene. It is unclear if and how the preview effect influences search guidance in subsequent searches. This is further obscured by the traditional approach of averaging response times across several subsequent trials. The influence of these initial activation processes on search efficiency in repeated search, particularly its influence on the SIE, has thus far not been investigated. 

Together with preliminary experimental evidence, the reviewed findings by Draschkow and colleagues [26] suggest that the activation of spatial priors might be the underlying cause and key ingredient to repeated search guidance and would be reflected in a prolonged first search compared to subsequent searches. This idea is further supported by Li and colleagues [37] who compared repeated search behavior on a computer and in a virtual environment. They stated that the most relevant factor for search guidance in their tested scenarios appeared to be learning of global spatial context. This resulted in more targeted fixations to important areas likely to contain the target. This is in line with the Visual Memory Theory of scene representation by Henderson and Hollingworth [38]. The theory posits that the allocation of attention leads to non-sensory, abstract visual representations of the scene, and the objects therein. These abstract representations include information such as object shape, position, and color and activate semantic knowledge [22,39,40,41]. The integration of those representations leads to a holistic representation of the entire scene, including spatial layout and scene categorization [21]. First, sensory input is processed as abstract representations into short-term memory, which then leads to the consolidation into long-term memory. The information can be maintained actively in short-term memory as well as retrieved from long-term memory and aids visual search and change detection [21,42]. According to the Visual Memory Theory [38], we would expect that spatial context first needs to be encoded through the allocation of attention and can then be used via maintenance and updating in short-term memory to activate spatial priors and guide subsequent search. 

While the reviewed literature has mainly investigated the mechanisms operating at the beginning of a repeated search episode in isolation, understanding ecological behavior requires to study these mechanisms as they unfold in natural behavior. The study presented here aimed to test the assumption that spatial prior activation causes an observable SIE, i.e., a substantial improvement in search time from the first to the second trial within the same scene. To this end, we conducted a repeated search study in a VR environment. To avoid ceiling effects, we used a controller-contingent window paradigm. Participants had the experience of searching in a dark room while holding a controllable flashlight which illuminated small parts of the environment, similar to a mouse-contingent window paradigm. VR provides powerful means for the investigation of natural behavior in a more ecologically valid setting. It enabled us to create naturalistic scenarios while maintaining a high level of control and to measure attentional allocation with a high temporal and spatial precision with the help of eye-tracking. Moreover, it has been shown that experiments in VR can enhance as well as reduce cognitive processes related to memory use compared to two-dimensional experimental settings [37,43]. Therefore, it is important to conduct experiments in a more realistic scenario such as VR to gain more insight into the generalizability of the results. 

To manipulate the activation of spatial priors, we created three different conditions: (1) participants either immediately had to start searching (Control), or (2) they had a preview of the scene and then started searching (Preview), or (3) they entered a gray waiting room for a brief period after the preview before starting to search to simulate a re-entrance of the room (Interruption). While the interruption should impede consolidation into long-term memory, the re-entrance after the interruption should enforce re-encoding and integration of the scene, causing the same processes upon searching as in the Control condition. Thus, we hypothesized that there would be an SIE in the Control and Interruption condition since in both conditions spatial priors would need to be activated at the start of the search, while we expected no effect in the Preview condition since the activation process already happened during the preview and did not need to be re-activated upon subsequent searching.

## 2. Materials and Methods

### 2.1. Data Availability and Preregistration

Experimental preprocessed data and the corresponding analysis script are available on the Open Science Framework at https://osf.io/5mncw/. The preregistration of the study is available at https://osf.io/xw5ug/. Statistical analyses were performed as preregistered if not mentioned otherwise. 

### 2.2. Participants

Thirty naïve German native speakers (21 female, mean age: 24.2 years, range: 18–41 years) were recruited at the Goethe University Frankfurt. The sample size was based on the rationale to have a fully balanced design. In addition, we conducted a preregistered power analysis to investigate how much power we have with 30 participants to detect the presence of our effect of interest. To this end, we simulated data and used an effect size based on a pilot study (the pilot study only included the Control condition and a completely different condition, i.e., no flashlight). With 30 participants, we have a power of more than 99% to detect an SIE in the Control condition, and 92% to detect an SIE one third the size of the Control condition. Three participants had to be excluded from further data analysis. In two cases, the experiment was aborted due to technical problems, and one participant aborted the experiment due to eye strain. The final sample consisted of 27 participants (21 female) with a mean age of 24 years (range: 18–41 years). All participants had normal or corrected-to-normal (contact lenses, no glasses) vision, were tested for visual acuity (at least 20/25) and normal color vision as assessed by the Ishihara test. All participants volunteered, gave informed consent, and were compensated with course credit or 8 €/h. The experimental procedure was approved by the local ethics committee of the Faculty of Psychology and Sport Sciences (2014-106R1) at Goethe University Frankfurt. 

### 2.3. Apparatus

Participants wore an HTC Vive head-mounted display (HMD; New Taipei City, Taiwan) equipped with a Tobii eye tracker (Tobii, Stockholm, Sweden) and held an HTC Vive controller in their dominant hand. The two 1080 × 1200 px OLED screens have a refresh rate of 90 Hz and a combined field of view of approximately 100° (horizontally) × 110° (vertically). The integrated Tobii eye tracker (Tobii, Stockholm, Sweden) recorded eye movements binocularly with a refresh rate of 120 Hz and a spatial accuracy below 1.1° within a 20° window centered in the viewports. The experiment was implemented in C# in the Unity 3D game engine (version 2017.3; Unity Technologies, San Francisco, CA, US) using SteamVR (version 1.10.26; Valve Corporation, Bellevue, WA, US) and Tobii eye-tracking software libraries (version 2.13.3, Tobii, Stockholm, Sweden) on a computer operated with Windows 10.

### 2.4. Stimuli

Fifteen virtual indoor scenes (three of each of five different room categories: bathroom, kitchen, living room, bedroom, office) were used in the experiment (see Figure 1). Every scene measured approximately 380 × 350 × 260 cm (length × width × height) in size, which was fitted to the actual room size of the laboratory to avoid any risk of walking into walls. Each scene consisted of eight global objects which are large, usually static objects (e.g., toilet, refrigerator, couch, bed, desk; also known as anchor objects, see [23]) and 20 local objects, which are smaller objects that are often interacted with (e.g., toothbrush, pan, remote control, alarm clock, keyboard). These local objects were used as target objects in the search task. An additional scene was used for practice trials, which was a gray room including ten objects, which were considered uncommon for typical living indoor spaces (e.g., hydrant, traffic light, stethoscope) to avoid any interference such as priming of any of the succeeding scenes. 

### 2.5. Experimental Design

For the visual search task, we implemented a controller-contingent window, intuitively similar to using a flashlight, which was considered to increase search difficulty by reducing the available visual information. The diameter of the flashlight window had a constant size of 8 degrees of visual angle. A pink circle was placed within the center of the flashlight window. Participants were instructed to place the circle on the target objects they had found and then pull the trigger button with their index finger on the backside of the controller. To investigate our hypotheses we created three different conditions: (1) participants had 10 s to explore the scene fully illuminated before the search trials with the controller-contingent window started (Preview condition), (2) participants had a 10 s preview just as in the Preview condition but it was followed by a 5 s long interruption in which participants were in a gray, empty room before the search started (Interruption condition), or (3) participants started searching immediately upon presentation of the scene (Control condition). Conditions were blocked and their order was balanced across participants. Every condition contained five randomly chosen rooms. Each room was from one of the five unique categories. The search order of objects was fully randomized across participants. Since our effect of interest was contained in the first trials, scenes switched after every four trials. Participants needed 60 trials to search through all scenes before starting over with the next four trials through the same scenes. This resulted in overall five scene visits for a total of 300 search trials. This setup created the possibility to investigate our effect of interest in 75 first trials per participant. Preview and interruption phases were repeated in the corresponding conditions. Each condition was preceded by four practice trials in the practice room before the first scene visit to demonstrate the procedure.

### 2.6. Procedure

Upon entering the lab, participants gave informed consent, performed both vision tests, and were familiarized with the HMD and how to use the controller. Next, the eye tracker was calibrated with a nine-point calibration grid. Subsequently, participants were instructed on the visual search task and how to navigate within the scenes with the flashlight attached to their controller. Participants were told to search as fast and precisely as possible, and were informed that they could move around the room while exploring and searching. Regarding preview and interruption phases, we informed participants that they could look around and were allowed to move freely while exploring. No information regarding strategies was given. Before entering a scene, participants were presented with an empty gray room with instructions written on a wall. Participants then had to position themselves on a blue square on the floor, which was the starting position for the scene and from where they could see most of the objects without obstructions. When the participants were ready, they pulled the trigger button to start the trials. Depending on the condition the scene belonged to, the procedure of entering a scene differed as can be seen in Figure 2. In the Preview condition, participants first had 10 s to freely explore the illuminated scene. The Interruption condition was additionally followed by a 5 s long presentation of a gray waiting room which looked like the instruction room but without any instructions. The Control condition had no preview and the search trials started immediately. At the beginning of a new block, participants were always informed via text written on a wall within an instruction room which block came next. When search trials started, the scene was entirely dark. A fixation cross appeared in the center of the participants’ visual field of view for 1 s, followed by the verbal target cue for 1.5 s. When the cue disappeared, the flashlight was turned on and participants had 30 s to find the target object. When having found the object, participants had to place the pink circle in the center of the flashlight illumination on the object and pull the trigger button with their index finger—similar to shooting it. In case the selected object was not the target or the timeout was reached, participants heard an error sound. Upon pulling the trigger or after the timeout, the flashlight was turned off and the fixation cross cueing the next search appeared. After four subsequent searches, participants again entered a gray room with instructions on the wall. The eye tracker was recalibrated after every block (i.e., after every 20 searches). Participants were allowed to take breaks before every eye tracker re-calibration if they wanted to. After successful completion of the experiment, participants were debriefed, and exploratively asked how they experienced the different conditions. The whole procedure lasted on average 1.25 h.

### 2.7. Data Analysis

Only correct trials were included in the analysis of response times. Trials, where the selected object was not the target or where the timeout was reached, were thus excluded (*N* = 649, 8.01%). We further excluded trials in which no saccades were detected (*N* = 237, 2.93%). This happened when participants already positioned themselves towards the target object while only the cue was visible. Fixations were determined based on the toolbox of the Salient360! Benchmark [44,45] using Python (version 3.7.1, Python Software Foundation, Delaware, US). By calculating the orthodromic distance and dividing it by the time difference, we obtained the velocity between gaze samples (°/ms) which was smoothed with a Savitzky–Golay filter [46]. Fixations were identified as filtered samples with a velocity of less than 120 °/ms. 

With the preprocessed data, we could further obtain our variables of interest, which were search initiation time (duration of the first fixation), time to first target fixation, decision time (response time—time to first target fixation), and gaze durations for each object before it became the target. The analyses were preregistered (see 2.1. Data availability and preregistration). To quantify the SIE, we averaged all correct trials #2–#4 of all 75 scene visits and subtracted them from the corresponding first trial. We performed this procedure for every variable of interest in the R statistical programming language (version 4.0.0, R Core Team, Vienna, Austria, [47]) using RStudio (version 1.2.1335, RStudio, Boston, MA, US, [48]). To establish whether an SIE is actually present in the different conditions, we calculated one-sample *t*-tests to test the mean against zero for each of them. Given that the SIE should always be positive (i.e., the first trial larger than the subsequent trials), we only expect the SIE to have positive values and therefore opted for one-sided tests. We further added Bayesian *t*-tests [49] to evaluate our data in terms of evidence for the null hypothesis. In accordance with Kass and Raftery [50], we interpreted the resulting Bayes factors (BF_10_) as either indicating evidence in favor of the alternative hypothesis (BF_10_ > 3), or indicating evidence in favor of the null hypothesis (BF_10_ < 0.33), or as inconclusive evidence (BF_10_ > 0.3 and BF_10_ < 3). Bayesian *t*-tests were computed with the BayesFactor package (version 0.9.12-4.2, [51]) and the default settings, i.e., with a Cauchy prior with a width of r = 0.707 centered on zero. We truncated the Cauchy prior to only allow for positive values which reflects our expectation of an SIE greater than zero [51]. 

We then calculated linear mixed models (LMMs) on the calculated SIEs using the lme4 package (version 1.1-23, [52]) to investigate our hypotheses. Performing a mixed-models approach allowed us to estimate both between-subject and between-stimulus variance simultaneously which is advantageous compared to traditional F1/F2 analyses of variance [53,54]. All LMMs were fitted with the maximum likelihood criterion. To predict the SIE in response times, we included search condition as a fixed effect and participant and scene as random effects. Please note that in the preregistration it was defined to include the target object as a random effect, however, this was not a sensible choice. Computed SIEs do not have a single target object because each SIE is calculated from four search trials. Thus, we included the scene as a random effect instead. We added summed gaze durations on the target object from previous trials and previews (aggregated like the SIE, i.e., gaze duration of the object of trial #1—average gaze duration of objects of trial #2 to trial #4) and the number of visits of the same scene as covariates with scaled and centered values to the fixed effects structure. Please note that gaze duration as a covariate was not preregistered and the results of the preregistered analysis can be found in Appendix A
Table A1. They are however essential, as viewing times of items predict memory performance [11,22,25] and could lead to a potential confound if not regressed out in the modeling approach. Further note, that we initially planned to calculate treatment contrasts but have decided that repeated contrasts provide a more suitable analysis to test our hypothesis [55]. The same LMMs were also calculated for the SIE based on search initiation time and time to first target fixation SIEs based on eye movements. Each model started with a maximal random effects structure [56] including random intercepts and slopes for participants and scenes. To avoid overparameterization and issues of non-convergence of such full models [57], we used principal component analysis (PCA) on the random-effects variance-covariance estimates to detect overparameterization. Random slopes that were not supported by the PCA and did not contribute significantly to the goodness of fit identified via likelihood ratio tests were removed. This procedure resulted in a model with a random-effects structure including intercepts for participants and scenes, but no random slopes. This was true for all LMMs, i.e., the behavioral SIE, the SIE on search initiation time, and the SIE on time to first target fixation. Figures were created with the ggplot2 package in R (version 3.3.0, [58]). Within-subject standard errors were calculated with the Rmisc package (version 1.5, [59]).

## 3. Results

### 3.1. Preregistered Analyses

#### 3.1.1. Search Initiation Effect in Response Time

There was a significant SIE in the Control condition, SIE = 1588 ms, SD = 1283, *t*(26) = 6.45, *p* < 0.001, BF_10_ = 44,470.58 (Figure 3). Participants did indeed search significantly longer in the first compared to the subsequent trials when the search started immediately upon entering the scene. The SIE in the Interruption condition was smaller but significant, SIE = 417 ms, SD = 1089, *t*(26) = 1.92, *p* = 0.033, BF_10_ = 1.91, however, the BF is deemed inconclusive as to whether an SIE was present or not. The Preview condition elicited no SIE, SIE = 41 ms, SD = 989, *t*(26) = 0.21, *p* = 0.419, BF_10_ = 0.24, which means that first searches were just as fast as subsequent searches. These findings go in line with our expectation of an SIE in the Control and no SIE in the Preview condition. However, we had expected a larger SIE in the Interruption condition. Comparing the conditions, the SIEs in the Interruption and Control condition differed significantly from each other, β = −1173.53, SE = 309.76, *t*(1769.49) = −3.79, *p* < 0.001, whereas contrary to our hypothesis the SIEs in the Preview and Interruption condition did not differ significantly, β = −337.96, SE = 312.47, *t*(1772.89) = −1.08, *p* = 0.280. Gaze durations significantly predicted SIEs, β = −645.77, SE = 127.42, *t*(1788.73) = −5.07, *p* < 0.001, but revisiting the scene played no role, β = −109.83, SE = 127.34, *t*(1770.81) = −0.86, *p* = 0.389. For a visualization of response time SIEs across scene revisits, see Figure A1 in the Appendix B, which illustrates that the process eliciting the SIE was triggered each time when entering the scene.

#### 3.1.2. Search Initiation Effect in Search Initiation Time and Time to First Target Fixation

To investigate a more fine-grained time course, we ran the same analyses on the SIE in search initiation time based on the first fixation duration as well as on the time to the first target fixation. The preregistered analyses without gaze duration as a covariate are in Table A2 and Table A3. Given that the SIE manifests in longer mental processes at the start, we would expect to also see a difference in search initiation time. However, we found inconclusive evidence for the SIE in the Control condition, SIE = 13 ms, SD = 34, *t*(26) = 1.57, *p* = 0.064, BF_10_ = 1.13, and evidence against SIEs in the Interruption, SIE = 1 ms, SD = 43, *t*(26) = 0.17, *p* = 0.434, BF_10_ = 0.235, and Preview condition, SIE = 3 ms, SD = 37, *t*(26) = 0.32, *p* = 0.378, BF_10_ = 0.26. In line with these results, there was neither a difference between the Interruption and Control condition, β = −11.24, SE = 10.92, *t*(1627.97) = −1.03, *p* = 0.304, nor between the Preview and Interrupt condition, β = 2.31, SE = 11.01, *t*(1753.88) = 0.21, *p* = 0.834, also shown in Figure 4a. While gaze durations did not predict search initiation time SIEs, β = −4.90, SE = 4.49, *t*(1786.73) = −1.09, *p* = 0.275, revisiting the scene significantly did so, β = 13.53, SE = 4.479, *t*(1760.27) = 3.02, *p* = 0.003.

Regarding time to first target fixation, we found an SIE in the Control condition, SIE = 1518 ms, SD = 1310, *t*(26) = 6.36, *p* < 0.001, BF_10_ = 35938, but just as in the response time SIEs no significant effect in the Interruption condition, SIE = 315 ms, SD = 987, *t*(26) = 1.58, *p* = 0.063, BF_10_ = 1.14, and in the Preview condition, SIE = 53 ms, SD = 941, *t*(26) = 0.29, *p* = 0.386, BF_10_ = 0.26. In line with the model of the response time SIE, Interruption and Control differed significantly from each other, β = −1227.22, SE = 291.23, *t*(1596.94) = −4.21, *p* < 0.001, but Preview and Interruption did not, β = −216.85, SE = 293.14, *t*(1740.43) = −0.74, *p* = 0.460, also see Figure 4b. These effects were not predicted by scene revisits, β = −103.77, SE = 119.54, *t*(1762.56) = −0.87, *p* = 0.386, but again by gaze durations, β = −640.48, SE = 119.58, *t*(1772.00) = −5.36, *p* < 0.001.

### 3.2. Exploratory Analyses

#### 3.2.1. Search Initiation Effect in Decision Time

In order to see if the process that drives the SIE affects more than one search phase, we decided to also look at the last phase of a search, i.e., the decision time. To do so, we again calculated one-sample one-sided *t*-tests and Bayes factors to establish the presence or absence of SIEs. Regarding decision time, there were no SIEs in the single conditions (i.e., Control, SIE = 43 ms, SD = 518, *t*(26) = 0.43, *p* = 0.334, BF_10_ = 0.29; Interruption, SIE = 107 ms, SD = 463, *t*(26) = 1.30, *p* = 0.103, BF_10_ = 0.77; Preview, SIE = −4 ms, SD = 443, *t*(26) = −0.05, *p* = 0.518, BF_10_ = 0.20). To compare conditions among each other, we used again an LMM as described earlier with conditions as fixed factors, participant and scene as random factors, and scene visit and gaze duration as fixed covariates. There were no differences between conditions as can be seen in Figure 4c, i.e., neither between Interruption and Control, β = 74.30, SE = 128.47, *t*(1625.60) = 0.58, *p* = 0.563, nor between Preview and Interruption, β = −111.89, SE = 129.07, *t*(1741.57) = −0.87, *p* = 0.386. And neither scene revisits nor gaze durations predicted decision time SIEs, i.e., β = 3.39, SE = 52.60, *t*(1742.00) = 0.06, *p* = 0.949, and β = −12.87, SE = 52.60, *t*(1769.31) = −0.25, *p* = 0.807, respectively. 

Taking these results into account, neither search initiation times nor decision times seem to be the locus of the SIE. 

#### 3.2.2. Incidental Memory

An explanation for the SIE could be that participants learned object locations during previous fixations [22,41]. As there were more opportunities to do so during previews in the Interruption and Preview condition but not in the Control condition, there is a potential confound. To investigate if the SIE is mainly driven by learning object locations, we selected only trials in which the target object has been fixated previously (either during trials, previews, or both, *N* = 5330, 73.88%). With this selection, object learning could potentially have happened for all selected trials. If we assume that the SIE is solely caused by object learning, we would expect to see no SIE at all regardless of the condition. But contrary to this hypothesis, there was still an SIE present in the Control condition, SIE = 1464 ms, SD = 1558, *t*(26) = 5.10, *p* < 0.001, BF_10_ = 1804.21. Now, there was also an SIE present in the Interruption condition roughly half the size the SIE of the Control condition, SIE = 784 ms, SD = 1271, *t*(26) = 3.06, *p* = 0.003, BF_10_ = 16.42, and no SIE in the Preview condition, SIE = 8 ms, SD = 1338, *t*(26) = 0.03, *p* = 0.488, BF_10_ = 0.21 (see Figure 5). Control and Interruption differed in their SIEs, β = −774.18, SE = 368.46, *t*(1208.99) = −2.10, *p* = 0.036, as well as Interruption and Preview, β = −732.58, SE = 341.34, *t*(1216.79) = −2.15, *p* = 0.032. The influence of revisiting the scene was not significant, β = −259.92, SE = 147.33, *t*(1219.48) = −1.76, *p* = 0.078, but gaze durations still predicted SIE significantly, β = −357.65, SE = 146.06, *t*(1226.43) = −2.45, *p* = 0.015. Therefore, it is rather unlikely that the SIE is purely caused by object learning during previews or search.

#### 3.2.3. Preview Effects beyond the First Search

We found a substantial difference in the overall response times between the Control and the two other conditions, which can be seen, e.g., in Figure 3a. To follow-up on this observation, we analyzed the overall response times as well as the improvement over time. We conducted an LMM predicting response times with trial number and condition, as well as their interaction. Gaze duration on the target object was added as a covariate. We further added random intercepts for subject and target object. Importantly, we used the Control condition as baseline (i.e., treatment contrasts) to analyze the benefit of both preview conditions. We found that compared to the Control condition the overall response times were reduced in the Interruption, β = −1524.23, SE = 121.51, *t*(6991.13) = −12.54, *p* < 0.001, as well as Preview condition, β = −1696.73, SE = 120.75, *t*(6965.18) = −14.05, *p* < 0.001. Moreover, the improvement in response times in the Control condition differed compared to both Interruption, β = 496.30, SE = 107.60, *t*(6979.14) = 4.61, *p* < 0.001, and Preview condition, β = 532.95, SE = 108.07, *t*(6974.42) = 4.93, *p* < 0.001. Follow-up LMMs with the same factor structure but for each individual search condition revealed that learning over time was largest in the Control condition, β = −625.49, SE = 85.41, *t*(2238.40) = −7.32, *p* < 0.001. The improvement over time was small but evident in the Interruption condition, β = −154.54, SE = 73.49, *t*(2299.81) = −2.10, *p* = 0.036, but no significant improvement was found in the Preview condition, β = −126.60, SE = 72.25, *t*(2263.60) = −1.75, *p* = 0.080. 

## 4. Discussion

In the study presented here, we replicated the anecdotally reported effect that the first search takes fundamentally longer than the subsequent searches in the same search environment. More importantly, we showed that this search initiation effect (SIE) found across various types of search scenarios (from letter displays to real-world scenes) generalizes also to realistic, navigable, and three-dimensional environments. To take a closer look during which search phase the SIE manifests, we recorded eye movements. Our results indicate that neither search initiation time nor decision time showed substantial SIEs. The locus of a strong SIE as particularly seen in response times in the Control condition seems to lie in attentional search guidance measured as the time to the first target fixation. We used three different preview conditions to test our hypothesis that this SIE is caused by an activation of spatial priors: (1) In the Control condition, participants immediately had to start searching, (2) in the Preview condition they had a 10 s long preview of the scene and then started searching, or (3) in the Interruption condition, they entered a gray waiting room for 5 s after the preview before starting to search. As expected, the effect was present in the Control condition in which spatial activation could only take place at the beginning of the search, while in the Preview condition, spatial activation was already possible during previews which were followed seamlessly by the searches. However, contrary to our predictions, we found no substantial SIE in the Interruption condition in which the preview of the scene and the searches were separated in time by an empty gray room. 

According to the Visual Memory Theory [38], the spatial layout is encoded—among other properties of the scene such as object positions and scene category—into short-term memory before being consolidated into long-term memory. While this encoding and consolidation process could already be initiated during the preview phases, these processes could only take place during the first search in the Control condition, thereby taking a toll on the search time of the first search as seen in the SIE. We implemented the Interruption condition to impede consolidation into long-term memory, inducing the need to re-encode and integrate the information of the actual scene just as in the Control condition, which should have manifested in a similar SIE despite a preview. Contrary to our predictions this was not the case. The SIE was much reduced in the Interruption condition compared to the Control condition. Obviously, the empty room did not impede consolidation processes due to the lack of interfering object information. Instead of impeding spatial activation needed to initiate search, the interruption could have worked as a time window for integration and consolidation of the just perceived information [29] or might have allowed participants to use memorization strategies during the interruption phase. The layout of the empty room could have even provided environmental support which can enhance spatial memory performance over time by facilitating spatial rehearsal [60,61,62]. This might have encouraged participants to project the scene from iconic memory into the empty room and maintain that memory representation thus reducing the necessity to reactivate the scene once the search started. Future studies will need to ensure that there is an actual interruption of maintained spatial visual information or a task that inhibits the active use of memorization strategies.

It could further be argued that participants simply learned the object locations during previews and that the SIE was therefore driven by specific object memory in contrast to the activation of more global spatial priors. Even though the difficulty of searching with a flashlight probably supported object memory acquisition [37], it is rather unlikely that object memory caused a reduction of the SIE due to several reasons. For one, memory for specific objects and their locations acquired through free viewing [63] or even active memorization [10] seems to be less useful for search compared to, for example, memory acquired through action-oriented behavior [12]. Even when participants have 60 s to memorize a scene for subsequent memory tasks, the benefit of doing so is minimal [37,63]. In our study, participants always had 10 s to explore the scene and did so five times (see Figure 2). While response times overall were improving with each revisit—suggesting successful generation and use of representations [21]—the SIE of the Control condition did not decrease over revisits (see Figure A1). This implies that searching through other scenes weakened these memory representations and a re-activation of spatial priors was necessary upon re-entering the scene. This could be because when searching other scenes in between, the spatial components needed for these searches interfered with the maintenance of the previously generated priors. If not interfered with, the activated spatial information could be maintained for several seconds [60]. This again speaks for the SIE marking the (re-)activation of global spatial priors in contrast to specific and local object memories. Consistent with this idea, some participants reported their impression that the previews became increasingly unnecessary throughout the experiment as they already felt as if they knew the scenes well enough, even in the Control condition where the scene was only ever seen through the beam of the flashlight. Finally, to rule out the possibility that object memories drove the SIE, we ran an exploratory analysis where we only inspected trials in which the target object was previously fixated (during previewing the scene and previous searches). Fixating objects leads to an encoding process, which is crucial to generate memory representations about location, identity, and the context they appear in [21,22,41,64]. These representations can be surprisingly stable and robust [42,65]. When only selecting trials with previous gaze durations on the target, we would expect to see no SIE in any condition because all objects were equally likely encoded and preserved in memory. However, contrary to that rationale we still found substantial SIEs in the Control and Interruption condition (see Figure 5). Therefore, it is unlikely that object memory alone caused the absence of SIEs in the Preview and Interruption condition.

Recording eye movements helped us to identify that the SIE was driven by the time to first target fixation while search initiation time and decision time did not contribute significantly. Therefore, activating spatial priors might make guidance less efficient during the first search, but once spatial priors are activated, they can be used to efficiently guide search again. This is in line with other research that found that various mental processes happen when encountering new visual information which are then used to facilitate behavior, such as gist processing [28,29,30,66] and scene categorization [31,32,33]. It is possible that even more mental processes feed into these activations and affect search performance, such as priming [67,68], activating scene grammar [3,25,69], and extracting summary statistics of spatial structure [70,71,72].

Finally, we also investigated post-hoc overall response time differences between the search conditions. Given the preview effect [21,28,29,30,35,36], we expected that the first search of the Control condition would be longer than the first searches of the other two preview conditions. This was indeed the case. However, response times in the Control condition never reached the level of the other two conditions. While we not only replicated the preview effect, we could further show that previews do not only benefit the very first search within a scene but have a profound and prolonged beneficial impact on the guidance of subsequent searches. Since the flashlight illumination during search was restricted to 8 degrees, participants in the Control condition never saw the scene as a whole, whereas the preview conditions provided global, spatial scene information at first glance. This global scene representation was able to then guide subsequent searches more efficiently than in the Control condition where a global scene representation would have to be incrementally established over the course of many searches with highly restricted visual inputs. Previous research on two-dimensional computer screens has found the useful field of view during search in scenes to be around approximately 8 degrees [73], but recent research has shown that the useful field of view in omnidirectional, three-dimensional environments likely goes beyond that [74]. This finding was further corroborated in this study, where restricting visual input in three-dimensional environments to 8 degrees of field of view obviously impeded performance in the Control condition from ever reaching the same level as in the two preview conditions. 

## 5. Conclusion

We showed that the SIE is a replicable phenomenon which can be manipulated in its strength. Our study suggests that upon visual processing of a scene, a process of activating spatial priors to enable orientation is initiated, which takes a toll on search time at first, but once activated it can be used to guide subsequent searches. Future research should specifically manipulate and disentangle the exact processes taking place during the activation of spatial priors and their maintenance in spatial working memory in order to better understand the processes that guide search behavior from the very outset. In our study, we employed a novel, realistic VR repeated search paradigm in a three-dimensional environment and made use of a state-of-the-art HMD equipped with an embedded eye tracker. Using VR allows researchers to investigate visual cognition in naturalistic scenarios while keeping full control over visual stimulation. Even more so, it allows to test scenarios which are tedious or even impossible to implement in real-world experiments. In our case, we tested participants’ visual search behavior in numerous different scenes while maintaining high control over the composition regarding scene category, room size, and number of global and local objects. In addition, the VR setup also allowed us to implement a flashlight paradigm that constantly restricted the visual input to an illuminated 8-degree window. We hope that our study inspires other vision researchers to implement new paradigms in VR which allow for rigorous and highly controlled investigations of daily human behavior in real-world scenarios.

## Figures and Tables

**Figure 1 brainsci-11-00044-f001:**
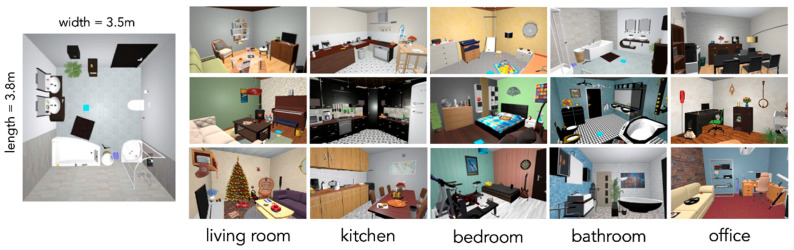
Bird’s-eye view of one of the bathrooms and one sample view of all the scenes that were used in the experiment. Blue squares indicate the starting position of the participants and were not visible during searching.

**Figure 2 brainsci-11-00044-f002:**
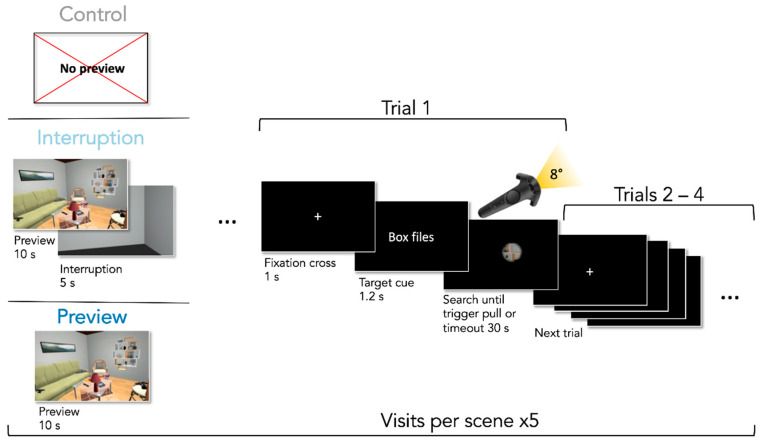
The procedure of the repeated visual search task. For visualization purposes, the cue is shown in English here but was presented in German in the original experiment. Cue and fixation cross are enlarged to increase legibility.

**Figure 3 brainsci-11-00044-f003:**
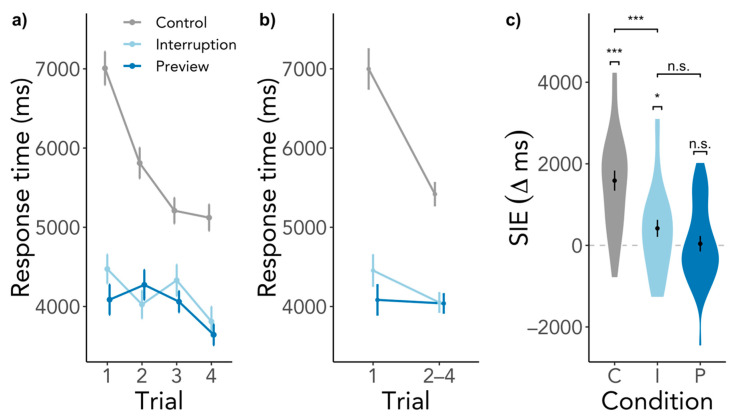
SIE in response times. (**a**) depicts the response times for all searches in a scene visit. (**b**) contains the same data as (**a**) but response times of the second to the fourth trial are shown as an average. (**c**) shows the difference between the response time of the first trial and the average of the second to the fourth trial. Higher values indicate a larger response time for the first trial than the subsequent trials, while the dashed line indicates the point of no difference in response times between trials. Capital letters indicate conditions, C = Control, I = Interruption, P = Preview. Error bars indicate within-subject standard error. n.s. = not significant, * *p* < 0.05, *** *p* < 0.001.

**Figure 4 brainsci-11-00044-f004:**
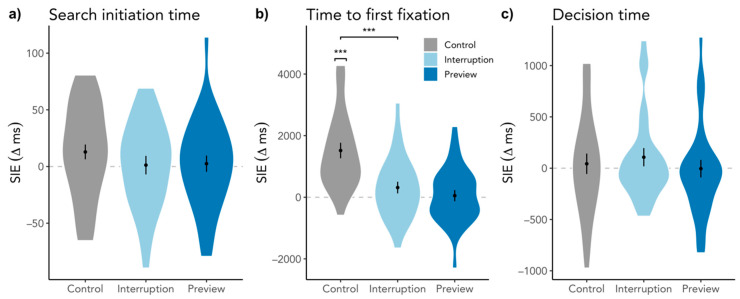
Difference between the time of the first trial and the average of the second to fourth trial for (**a**) search initiation time, (**b**) time to first target fixation, and (**c**) decision time. Plots are ordered in their chronological appearance during searching. Higher values indicate a larger response time for the first trial than the subsequent trials, while the dashed line indicates the point of no difference in response times between trials. Error bars indicate within-subject standard error. For reading purposes, we only denoted significant comparisons. *** *p* < 0.001.

**Figure 5 brainsci-11-00044-f005:**
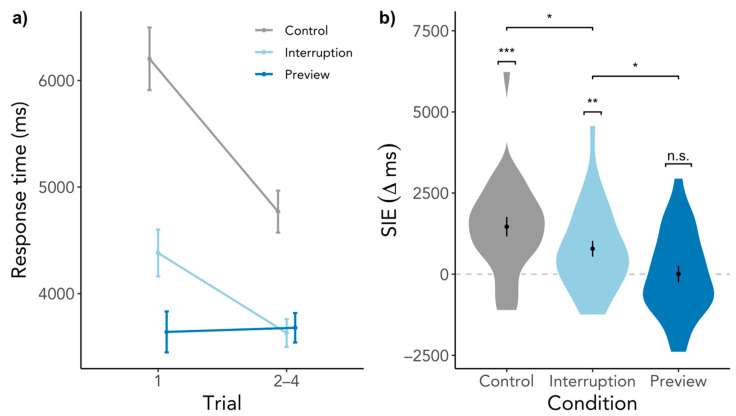
SIE in response times only for trials in which the target object has been fixated previously. (**a**) depicts the response times for the first trial in a scene visit and the average of the response times of the second to the fourth trial. (**b**) shows the difference between the response time of the first trial and the average of the second to the fourth trial. Higher values indicate a larger response time for the first trial than the subsequent trials, while the dashed line indicates the point of no difference in response times between trials. Error bars indicate within-subject standard error. n.s. = not significant, * *p* < 0.05., ** *p* < 0.01, *** *p* < 0.001.

## Data Availability

The here presented data is publicly available in a preprocessed format as well as the analysis script which computes all here presented analyses. Data and script are available at https://osf.io/5mncw/.

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
