# Peer review of "Get Your Guidance Going: Investigating the Activation of Spatial Priors for Efficient Search in Virtual Reality"

_brainsci, 2021, doi:10.3390/brainsci11010044_

Round 1
Reviewer 1 Report
This manuscript describes one study examining the effect of previewing a scene on search times in a virtual reality paradigm. The main finding is that the first search within a block of searches in the same scene was slower than subsequent searches in the control condition, but there was no slowing on the first trial if a preview was permitted.
Overall, this study seems quite solid and well in line with other work in this field. As such, I don't believe that it will be difficult to address the few comments that I have.
- The primary question I am left with after reading is why is the control condition still slower on search 5? That it is slower on search 1 is well-discussed, but I would like more discussion on why that first-search effect lingers to other, subsequent searches. As part of this discussion, consider whether Figure A1 makes sense to be in the main body of the manuscript.
- Similarly, I found figure A2 to be very interesting and think it might merit discussion in section 3.2.2
- A point of confusion I had while reading the abstract and introduction was what was meant by "first" search. I was unsure if it was the first search of the whole experiment, first search after a break, first search for a given target, etc. This was described clearly in the methods, but more clarity earlier would help to situate the reader.
- P.3 L.38 HDM should be HMD
- Consider moving section 2.6 earlier, such as at the very beginning of the methods. There is discussion of things being preregistered before the preregistration is noted.
Author Response
We thank the reviewer for the thoughtful comments and constructive, precise feedback, which helped to improve our manuscript. We indicated below how we addressed the reviewer's concerns. Quotes of the review are in italic.
- The primary question I am left with after reading is why is the control condition still slower on search 5? That it is slower on search 1 is well-discussed, but I would like more discussion on why that first-search effect lingers to other, subsequent searches. As part of this discussion, consider whether Figure A1 makes sense to be in the main body of the manuscript.
Reply: Thanks, this indeed is a very interesting question and one that was addressed by Reviewer 2 as well. We did not have a strong a priori hypothesis about an overall search time difference of the Control condition compared to the other two conditions, but we agree with Reviewer 2 that the most likely reason for this effect was a more profound than expected effect of the previews. We added Figure A1 as panel a) to Figure 3 in the main text, reported an additional analysis concerning these response time differences in the new section “3.2.3. Preview effects beyond the first search” and added a paragraph about this in the discussion.
- Similarly, I found figure A2 to be very interesting and think it might merit discussion in section 3.2.2
Reply: We agree with the reviewer, it is certainly interesting that the SIE especially in the Control condition remains constant, indicating that the process eliciting the SIE is triggered each time entering the scene. We now refer to Figure A1 (its new name) in the Results section, and deliberate on these findings in the Discussion (see page 12, lines 468 – 480).
- A point of confusion I had while reading the abstract and introduction was what was meant by "first" search. I was unsure if it was the first search of the whole experiment, first search after a break, first search for a given target, etc. This was described clearly in the methods, but more clarity earlier would help to situate the reader.
Reply: Thanks for pointing out this reason for possible confusions. We made some additions to the abstract and the introduction and hope that it is clearer now.
- P 3 L.38 HDM should be HMD
Reply: Good catch! We corrected it.
- Consider moving section 2.6 earlier, such as at the very beginning of the methods. There is discussion of things being preregistered before the preregistration is noted.
Reply: We followed the reviewer’s suggestion and moved the section concerning preregistration to the very beginning of the methods section, now being “2.1. Data availability and preregistration”.
Reviewer 2 Report
This study uses VR to study visual search, focusing on the preview-effect. Search speed is compared between three conditions: after a 10s preview of the scene, after a 10s preview of the scene followed by a 5s interruption, and without preview. Results show a large preview effect: search was faster in the two preview conditions than in the control condition. This difference reduced across trials.
General evaluation: the method is innovative. The results are perhaps not quite as informative.
Comments:
-It took me a while to understand what the question was. The first paragraph is very broad, discussing influence of prior knowledge in guiding behaviour. The Introduction then turns to the use of memory in search, followed by a more detailed discussion of the SEI. It would help if the objective/topic of the study was made clear early on. Specifically, I suggest introducing the preview effect early on and discussing this effect in some detail.
-The study is introduced in the context of the “search initiation effect”. How is this different from the preview effect? That is, in the Control condition, the first trial effectively serves as a preview for subsequent trials; or, put differently, in the Preview condition, the preview is like a first trial. The SEI should be more fully integrated with the preview effect, as these partly reflect the same process (e.g., activation of spatial priors, familiarity with the scene). Currently, the paper suggests that the SEI is an independent effect that has been ignored in the literature.
-Related to the above comment: I would find it more informative to see the results for the 4 trials separately as main analysis (as in Appendix B) and analyze the improvement across time for the different conditions. The three conditions clearly show different time courses. These time courses are more informative, for example in showing that the Control condition is still slower than the Preview/Interruption conditions on Trial 4 (and Trial 4 is still slower than Trial 1 in the Preview/Interruption conditions!). This suggests that the fully illuminated scene preview provided more information than the information revealed during the spotlight search trials. This could be discussed.
-“The aim of the study presented here was to test the assumption that spatial prior activation causes an observable SIE, i.e., a substantial improvement in search time from the first to the second trial.” This made me wonder: What is the alternative hypothesis? Is there an account that predicts no difference from first to second trial?
-The Interruption condition could be better motivated: why was it included? What was the hypothesis for this condition? This is discussed later on (In Discussion) but would also need to be covered in the Introduction: “We implemented the Interruption condition in order to impede consolidation into long-term memory, inducing the need to re-encode and integrate the information of the actual scene just as in the Control condition, which should have manifested in a similar SIE despite a preview“
-As the authors discuss, there are many reasons for finding faster search after a 10s preview, with less room for improvement in subsequent trials (gist processing, scene categorization, priming, activating scene grammar, extracting summary statistics of spatial structure). Could the authors be clearer about what we have learned from this study? As the method is very promising and nicely implemented, perhaps a stronger focus on the method may better highlight the novel contribution of the study. This could also include a review of other visual search studies using VR and, more generally, a discussion of the benefits of using VR – why/how might it reveal differences relative to search in pictures?
Author Response
We thank the reviewer very much for the constructive feedback on our manuscript. The comments were very helpful, inspiring and improved our manuscript a lot. We indicated below how we addressed the concerns. Quotes of the review are in italic.
- It took me a while to understand what the question was. The first paragraph is very broad, discussing influence of prior knowledge in guiding behaviour. The Introduction then turns to the use of memory in search, followed by a more detailed discussion of the SEI. It would help if the objective/topic of the study was made clear early on. Specifically, I suggest introducing the preview effect early on and discussing this effect in some detail.
Reply: We are thankful for the reviewer’s suggestion. We added more information on the preview benefit and introduce our research question early on. We hope this frames our introduction a bit better now.
- The study is introduced in the context of the “search initiation effect”. How is this different from the preview effect? That is, in the Control condition, the first trial effectively serves as a preview for subsequent trials; or, put differently, in the Preview condition, the preview is like a first trial. The SEI should be more fully integrated with the preview effect, as these partly reflect the same process (e.g., activation of spatial priors, familiarity with the scene). Currently, the paper suggests that the SEI is an independent effect that has been ignored in the literature.
Reply: We appreciate the reviewer’s thoughtful comment and apologize for being too ambiguous on this. Indeed, the preview effect as well as the search initiation effect (SIE) most likely involve similar processes. Our group has studied the preview effect on search extensively (see Võ & Henderson, 2010, 2011; Võ & Schneider, 2010). However, preview effect studies typically isolate the preview from subsequent searches, whereas here we were interested to study the SIE (if present) as a part of the integrated process of searching. That is, the preview effect focuses on how presenting some information (e.g., spatial layout) first, affects some process afterwards (e.g., search). Here, we focused on understanding this mechanism when it is integrated within the enacted process (in this case searching). The Control condition here exemplifies a normal search scenario, similar to cases in which substantial SIEs were visible, but never addressed (usually due to averaging), in previous studies. Here, we were interested in the nature of these observed SIEs and whether they can be modulated by our preview manipulations, since this provides new insights as to what type of information processing within the first glimpse of a scene actually creates the SIE. So, we absolutely agree with the reviewer that the activation of spatial priors (as known from Preview manipulations) are the major explanatory factor of the SIE and hope that we now provide the experimental evidence to support this claim. We further added more information about this relationship to the introduction.
- Related to the above comment: I would find it more informative to see the results for the 4 trials separately as main analysis (as in Appendix B) and analyze the improvement across time for the different conditions. The three conditions clearly show different time courses. These time courses are more informative, for example in showing that the Control condition is still slower than the Preview/Interruption conditions on Trial 4 (and Trial 4 is still slower than Trial 1 in the Preview/Interruption conditions!). This suggests that the fully illuminated scene preview provided more information than the information revealed during the spotlight search trials. This could be discussed.
Reply: We thank the reviewer for this suggestion, which aligns closely with a request from Reviewer 1 and overall improved our manuscript substantially. We agree that the overall response time differences between the Control and the other two conditions are indeed quite interesting and as we had no strong a priory hypothesis about it, were rather unexpected. The previews seemed to induce a boost that affected all four searches, and as the reviewer points out, went beyond the information gathered in the spotlight search trials. We performed analyses on the time course ands overall response times differences (see new section “3.2.3. Preview effects beyond the first search”) and elaborated on this in the discussion. Thanks for highlighting this important point, which would have slipped through our analysis.
- “The aim of the study presented here was to test the assumption that spatial prior activation causes an observable SIE, i.e., a substantial improvement in search time from the first to the second trial.” This made me wonder: What is the alternative hypothesis? Is there an account that predicts no difference from first to second trial?
Reply: This is an important point, which we might have not made clear enough in the manuscript. Critically, we see that in most of the repeated search literature the majority of learning (i.e., improvement in search time) seems to happen early-on – within the first two searches of the display. This rather non-linear learning is usually ignored or averaged out. So the question rather is if there is an account that explains the “major” search time improvement – when compared to the less dramatic improvements in further searches. Here we were interested in what actually drives this early learning. In line with the reviewer’s arguments, we believe spatial prior activation to play not only an important role, but a role which explains far more “learning” variance compared to other factors (e.g., object learning or incidental memory). We expected this to be the case from what we know from the preview effect literature and here we could provide evidence for this interpretation within a framework of natural actions, such as ongoing repeated search. In line with the reviewer’s comment 6, we hope that future studies can address the contributions of these spatial activation prior in a more granular fashion – following the rich tradition in the preview effect literature but applying it to actual continuous and uninterrupted behavior.
- The Interruption condition could be better motivated: why was it included? What was the hypothesis for this condition? This is discussed later on (In Discussion) but would also need to be covered in the Introduction: “We implemented the Interruption condition in order to impede consolidation into long-term memory, inducing the need to re-encode and integrate the information of the actual scene just as in the Control condition, which should have manifested in a similar SIE despite a preview“
Reply: We thank the reviewer for pointing out this missing information in the introduction. We added our reasoning for the Interruption condition to the introduction.
- As the authors discuss, there are many reasons for finding faster search after a 10s preview, with less room for improvement in subsequent trials (gist processing, scene categorization, priming, activating scene grammar, extracting summary statistics of spatial structure). Could the authors be clearer about what we have learned from this study? As the method is very promising and nicely implemented, perhaps a stronger focus on the method may better highlight the novel contribution of the study. This could also include a review of other visual search studies using VR and, more generally, a discussion of the benefits of using VR – why/how might it reveal differences relative to search in pictures?
Reply: We thank the reviewer for the compliment on our methodological implementation. In line with the reviewer, we now further highlight the technical advances of our study in the discussion. We further believe that our study adds new theoretical insights to the understanding of natural search behavior. While future studies will address the contributions of the different spatial activation factors outlined by the reviewer more granularly, we could demonstrate that the activation of spatial priors in general slowdown search time at first but guide subsequent searches for a prolonged, task-relevant period.
Round 2
Reviewer 2 Report
The authors have addressed my concerns.